# Learnable Invisible Backdoor for Diffusion Models

## Abstract

Diffusion models have shown tremendous potential for high-quality image generation in recent years. Accordingly, there has been a rising focus on security threats associated with diffusion models, primarily because of their potential for malicious utilization. Recent studies have shown diffusion models are vulnerable to backdoor attack, which can make diffusion models generate designated target images given corresponding triggers. However, current backdoor attacks depend on manually designed trigger generation functions, which are usually visible patterns added to input noise, making them easily detected by human inspection. In this paper, we propose a novel and general optimization framework to learn invisible trigger, making the inserted backdoor more stealthy and robust. Our proposed framework can be applied to both unconditional and conditional diffusion models. In addition, for conditional diffusion models, we are the first to show how to backdoor diffusion models in text-guided image editing/inpainting pipeline. Extensive experiments on various commonly used samplers and datasets verify the effectiveness and stealthiness of the proposed framework. Our code is publicly available on `https://github.com/invisibleTriggerDiffusion/invisible_triggers_for_diffusion`.

## 1 Introduction

Recently, diffusion models have demonstrated high-quality and diverse image generation capability (Ho et al., 2020; Nichol & Dhariwal, 2021; Dhariwal & Nichol, 2021; Ho & Salimans, 2022; Nichol et al., 2021). Based on diffusion models, many popular applications have been developed including GLIDE (Nichol et al., 2021), Imagen (Saharia et al., 2022), and Stable Diffusion (Rombach et al., 2022), which provide a great tool for creative content generation. By slowly adding random noise to data, diffusion models learn to reverse the diffusion process to construct desired data samples from the noise. However, this procedure also brings new security threats. Salman et al. (2023) manages to conduct adversarial attack with diffusion models and uses them to prevent images from malicious editing. They proposes to use adversarial examples instead of original images to create unrealistic images after editing. Also, backdoor attacks have also been studied in diffusion models (Chou et al., 2023a; Chen et al., 2023b; Chou et al., 2023b; Struppek et al., 2022), showing the potential threat caused by backdooring the diffusion procedure. However, they either inject visible image triggers and additional text into input noise/prompt to backdoor diffusion models (Chou et al., 2023a; Chen et al., 2023b; Chou et al., 2023b), or replace input text characters with non-Latin characters to backdoor only text encoder, instead of diffusion models (Struppek et al., 2022). Albeit achieving high success rate, triggers in Chou et al. (2023a) and Chen et al. (2023b) can be easily detected by human inspection. To the best of our knowledge, none of the previous works have considered invisible image triggers for backdooring diffusion models, which means that the security threat posed by backdoor attacks has not been fully explored. For more detailed discussion on the importance and motivation of backdooring diffusion models with invisible triggers, please refer to Appendix B.

In this paper, as shown in Figure 1, we explore backdoor attacks with invisible image triggers for both unconditional and conditional diffusion models. Specifically, we propose a novel and general framework formulated by bi-level optimization to learn (input-aware) invisible triggers. In the inner optimization, given the fixed diffusion model, a trigger generator is optimized to make the diffusion model generate target image when sampled with the generated trigger and, at the same time, keep the

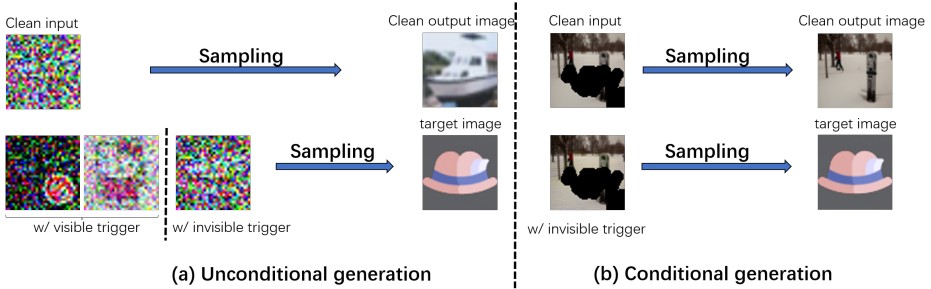

Figure 1: Illustration for our proposed invisible triggers, and visible triggers used in Chou et al. (2023a;b); Chen et al. (2023b).

trigger invisible. In the outer optimization, given the fixed trigger generator, the diffusion model is optimized to achieve a good performance on both clean data and poisoned data. Therefore, under our framework, backdooring on conditional and unconditional diffusion model could be incorporated into one optimization problem. Note that although previous works (Doan et al., 2021b;a) also utilize a bi-level optimization to achieve invisible triggers, learning an invisible backdoor trigger in the context of diffusion models is totally different than finding one in the classification model. For detailed discussion, please refer to Appendix C. We also manage to derive a general loss function to handle a series of efficient samplers such as DDIM (Song et al., 2020) and DPMSolver (Lu et al., 2022) instead of only one sampler as in Chou et al. (2023a). We then derive the loss function accordingly given different priors. Specifically, when the prior is a random Gaussian noise for the unconditional diffusion model, we train the diffusion model to learn different distribution mappings so that the diffusion model would generate the target image given an initial noise that contains the trigger. We show the proposed framework are able to insert more than one trigger-target pair and generate a instance adaptive initial noise rather than a uniform perturbation. For the conditional diffusion model, by learning a simple neural network to generate instance-adaptive and invisible trigger, we manage to insert backdoor into the the additional priors so that the conditional diffusion model would generate the target image regardless of any given text. While previous works focus on designing the trigger in the text domain, we are the first to successfully plant the invisible backdoor into the image space.

To summarize, for the first time, we propose a novel and general optimization framework to inject (input-aware) invisible image triggers into both unconditional and conditional diffusion models, which makes the injected backdoor more stealthy and robust. Specifically, for the unconditional generation, we also derive a general loss function which makes the trained diffusion models applicable to a wide range of commonly used samplers. On the other hand, for the conditional generation, we are also the first to show how to backdoor conditional diffusion models in the text-guided image editing/inpainting pipeline. We conduct extensive experiments on various datasets and samplers to show the effectiveness of the proposed framework.

## 2 RELATED WORK

**Diffusion models** As a new family of powerful generative models, diffusion models could achieve superb performance on high-quality image synthesis (Ho et al., 2020; Nichol et al., 2021; Saharia et al., 2022; Rombach et al., 2022). They have shown impressive results on various tasks, such as class-to-image generation (Dhariwal & Nichol, 2021; Ho & Salimans, 2022), text-to-image generation (Saharia et al., 2022), image-to-image translation (Meng et al., 2021), text-guided image editing/inpainting (Nichol et al., 2021; Rombach et al., 2022), and so on. With the powerful capability, the research community has started to focus on the potential security issues that diffusion models may introduce. In this paper, we propose a strong attack framework which can make diffusion models perform maliciously when some invisible patterns are injected into the input, revealing the potential severe security risk that previous works did not cover.

**Backdoor attacks on diffusion models** Recently, diffusion models have been shown to be vulnerable to backdoor attacks (Chou et al., 2023a; Chen et al., 2023b; Chou et al., 2023b; Struppek

et al., 2022). Struppek et al. (2022) proposed to backdoor the text encoder only in text-to-image diffusion models by replacing text characters with non-Latin characters, showing the potential threat in text-to-image generation. However, their method did not consider the backdoor threat on the diffusion models, limiting the practical use of the method. Chou et al. (2023a) and Chen et al. (2023b) proposed to backdoor diffusion models in different ways. Their methods focus on backdooring unconditional diffusion models, which may not be applicable to backdoor diffusion models in conditional case. Very recently, Chou et al. (2023b) proposed a unified framework on backdoor attack for both unconditional diffusion models and text-to-image diffusion models. However, to the best of our knowledge, none of previous works consider invisible triggers for backdoor attack in diffusion models. In this paper, we propose a novel and general framework to inject invisible triggers into both unconditional and conditional diffusion models. Moreover, we are also the first to show how to backdoor conditional diffusion models in text-guided image editing/inpainting pipeline.

## 3 METHODOLOGY

### 3.1 THREAT MODEL

We consider a similar threat model to previous work (Chou et al., 2023a;b) where there are two parties in the attack scenario. An *attacker* injects backdoors into diffusion models then releases backdoored models and a *user* downloads pre-trained models from the web for further usage so that the *user* can have full access to backdoored models. Also, the *user* will have a subset of clean data to evaluate the performance of the models. The attacker is allowed to control the training procedure of the diffusion including the training or fine-tuning. Also, modification on the training datasets is allowed so that the attacker is capable to add some additional examples into training dataset. Through the training, the attacker aims to release backdoored models that perform designated behavior when the input is injected with the trigger and behave normally on inputs without the trigger. Hence the attack goals are achieving high utility and high specificity of backdoored models. High utility means the backdoored models should perform similarly to clean models and generate high quality images that follow the training dataset distribution; high specificity means that the backdoored models should perform maliciously by generating target images once the trigger is in the initial noise or inputs. Since current backdoor triggers all make poisoned images visually different from clean images and universal across all the inputs, they can be detected easily by universal perturbation-based detection and human inspection. In this paper, we aims to learn input-aware invisible triggers to make the injected backdoor stronger and more stealthy.

### 3.2 PRELIMINARY ON DIFFUSION MODELS

Diffusion models consist of two processes, the forward/diffusion process as a Markov chain and the backward/reverse process (Ho et al., 2020). In the diffusion process, given an image sampled from real data distribution, noise randomly sampled from Gaussian distribution is gradually added to the image until it is transformed into noise. Specifically, Gaussian noise is gradually added to real data $\boldsymbol{x}_0 \in \mathbb{R}^d$ sampled from the real data distribution $q(\boldsymbol{x}_0)$ for $T$ steps, producing a series of noisy copies $\boldsymbol{x}_1, \boldsymbol{x}_2, \cdots, \boldsymbol{x}_T \in \mathbb{R}^d$. As $T \to \infty$, $\boldsymbol{x}_T$ will follow the isotropic Gaussian distribution, i.e., $\boldsymbol{x}_T \sim \mathcal{N}(\boldsymbol{0}, \boldsymbol{I})$. More formally, the diffusion process is defined as

$$q(\boldsymbol{x}_{1:T}|\boldsymbol{x}_0) := \prod_{t=1}^{T} q(\boldsymbol{x}_t|\boldsymbol{x}_{t-1}), \qquad q(\boldsymbol{x}_t|\boldsymbol{x}_{t-1}) := \mathcal{N}(\boldsymbol{x}_t; \sqrt{1-\beta_t}\boldsymbol{x}_{t-1}, \beta_t\boldsymbol{I}). \tag{1}$$

By defining $\alpha_t := 1 - \beta_t$, $\bar{\alpha}_t := \prod_{s=1}^{t} \alpha_s$, we have

$$q(\boldsymbol{x}_t|\boldsymbol{x}_0) = \mathcal{N}(\boldsymbol{x}_t; \sqrt{\bar{\alpha}_t}\boldsymbol{x}_0, (1-\bar{\alpha}_t)\boldsymbol{I}). \tag{2}$$

We can simulate the true data distribution by reversing the diffusion process described above. Hence the reverse process can also be defined as a Markov chain with learned Gaussian transitions starting from $p(\boldsymbol{x}_T) = \mathcal{N}(\boldsymbol{x}_T; \boldsymbol{0}, \boldsymbol{I})$:

$$p_\theta(\boldsymbol{x}_{0:T}) := p(\boldsymbol{x}_T) \prod_{t=1}^{T} p_\theta(\boldsymbol{x}_{t-1}|\boldsymbol{x}_t), \qquad p_\theta(\boldsymbol{x}_{t-1}|\boldsymbol{x}_t) := \mathcal{N}(\boldsymbol{x}_{t-1}; \boldsymbol{\mu}_\theta(\boldsymbol{x}_t, t), \boldsymbol{\Sigma}_\theta(\boldsymbol{x}_t, t)). \tag{3}$$

The training is performed by optimizing the variational lower bound, which can be further rewritten as comparing the KL divergence between the $p_\theta(\boldsymbol{x}_{t-1}|\boldsymbol{x}_t)$ and $q(\boldsymbol{x}_{t-1}|\boldsymbol{x}_t, \boldsymbol{x}_0)$ as

$$q(\boldsymbol{x}_{t-1}|\boldsymbol{x}_t, \boldsymbol{x}_0) = \mathcal{N}(\boldsymbol{x}_{t-1}; \tilde{\boldsymbol{\mu}}_t(\boldsymbol{x}_t, \boldsymbol{x}_0), \tilde{\beta}_t \boldsymbol{I}), \qquad \tilde{\boldsymbol{\mu}}_t(\boldsymbol{x}_t, \boldsymbol{x}_0) = \frac{1}{\sqrt{\alpha_t}} \left( \boldsymbol{x}_t(\boldsymbol{x}_0, \boldsymbol{\epsilon}) - \frac{\beta_t}{\sqrt{1-\bar{\alpha}_t}} \boldsymbol{\epsilon} \right), \tag{4}$$

where $\boldsymbol{x}_t(\boldsymbol{x}_0, \boldsymbol{\epsilon}) = \sqrt{\bar{\alpha}_t}\boldsymbol{x}_0 + \sqrt{1-\bar{\alpha}_t}\boldsymbol{\epsilon}, \boldsymbol{\epsilon} \sim \mathcal{N}(\boldsymbol{0}, \boldsymbol{I})$.

Due to the property of Gaussian distribution, the loss function can be further written as

$$L = \mathbb{E}_{t,\boldsymbol{x}_0,\boldsymbol{\epsilon}} \left[ \|\boldsymbol{\epsilon} - \boldsymbol{\epsilon}_\theta(\sqrt{\bar{\alpha}_t}\boldsymbol{x}_0 + \sqrt{1-\bar{\alpha}_t}\boldsymbol{\epsilon}, t)\|^2 \right]. \tag{5}$$

### 3.3 Optimization framework for learnable invisible trigger

In this paper, we aim to inject invisible image triggers into both unconditional and conditional diffusion models to make the backdoor more stealthy and effective. To this end, we formulate the problem of injecting an invisible trigger in a bi-level optimization framework, which learns the invisible trigger to be inserted given different priors. For the purpose of generality, let $g$ be a trigger generator that generates invisible triggers given different priors $P$, $\mathcal{A}$ be the trigger insertion function which inserts the invisible trigger generated by $g$ into $P$, and $\boldsymbol{y}$ be the target image that the backdoored diffusion model will generate when the trigger is activated. Let $\boldsymbol{\epsilon}_\theta$ be the diffusion model which takes the prior $P$ as input to predict the noise, and $\mathcal{S}$ denote the whole sampling process of diffusion models which takes diffusion model $\boldsymbol{\epsilon}_\theta$ and $P$ as input to generate real data by iterative sampling. We first show the general optimization framework for different priors (i.e., unconditional and conditional generation). Then we will elaborate on the specific parameterization of $g$ and $\mathcal{A}$ for different priors $P$ in detail.

We formulate generating invisible backdoor for difussion model problem into a bi-level optimization. In the inner optimization, we optimize the trigger generator $g$ given fixed $\boldsymbol{\epsilon}_\theta$ to generate the target image $\boldsymbol{y}$. The MSE(mean squared error) is used as loss function to train $g$ as:

$$L_{inner}(\boldsymbol{\epsilon}_\theta, P, g(P)) = \left\| \mathcal{S}\Big(\boldsymbol{\epsilon}_\theta, \mathcal{A}(P, g(P))\Big) - \boldsymbol{y} \right\|^2. \tag{6}$$

At the same time, to ensure the invisibility of the generated trigger, the generated trigger is bounded by $\ell_p$ norm ($\ell_\infty$ used in this paper), where we use PGD (Madry et al., 2017) optimization to force the constraint. To optimize $g$, all intermediate results during sampling have to be stored to compute the gradient with respect to $g$. Hence it becomes infeasible to sample many steps like the original DDPM (Ho et al., 2020) sampling. Alternatively, we use DDIM (Song et al., 2020) to perform the accelerated sampling process to make the optimization tractable.

For the outer optimization of bi-level optimization, we optimize the diffusion model $\boldsymbol{\epsilon}_\theta$ to correctly predict the noise for both clean data and poisoned data (i.e., backdoor diffusion process). Let $L_{outer}$ denote the loss function for the outer optimization, and details for $L_{outer}$ given different priors $P$ will be described later. Hence the loss function for outer optimization can be written as

$$L_{outer}\left(\boldsymbol{\epsilon}_\theta, \boldsymbol{x}_0, P, g(P), \boldsymbol{y}, t\right). \tag{7}$$

Hence the whole bi-level optimization framework for learnable invisible trigger can be written as

$$\min_\theta L_{outer}\left(\boldsymbol{\epsilon}_\theta, \boldsymbol{x}_0, P, g_\theta^*(P), \boldsymbol{y}, t\right)$$
$$\text{s.t.} \quad g_\theta^* = \arg\min_g \left\| \mathcal{S}\Big(\boldsymbol{\epsilon}_\theta, \mathcal{A}(P, g(P))\Big) - \boldsymbol{y} \right\|^2, \ \|g(P)\|_\infty \le C, \tag{8}$$

where $\|\cdot\|_\infty$ denotes the $\ell_\infty$ norm, and $C$ is the norm bound of the trigger. Given the general framework in Equation 8, we now describe the specific parameterization of each term in the framework for different priors.

#### 3.3.1 Backdooring unconditional diffusion model

When we backdoor unconditional diffusion model, the prior could be regarded as random Gaussian noise $P = \boldsymbol{\epsilon}, \boldsymbol{\epsilon} \sim \mathcal{N}(\boldsymbol{0}, \boldsymbol{I})$. Specifically, we optimize $g$ as a universal trigger generator for any

random noise sampled from $\mathcal{N}(\mathbf{0}, \boldsymbol{I})$. That is, $g(\boldsymbol{\epsilon}) = \boldsymbol{\delta}, \forall \boldsymbol{\epsilon} \sim \mathcal{N}(\mathbf{0}, \boldsymbol{I})$. The trigger injection function $\mathcal{A}$ then could be defined as $\mathcal{A}(P, g(P)) = \mathcal{A}(\boldsymbol{\epsilon}, \boldsymbol{\delta}) = \boldsymbol{\delta} + \boldsymbol{\epsilon}$. In other words, we aim to make the diffusion model generate the target image given any poisoned noise sampled from $\mathcal{N}(\boldsymbol{\delta}, \boldsymbol{I})$. By generating different $\boldsymbol{\delta}$, our proposed method could insert multiple invisible universal trigger-target pairs simultaneously. Furthermore, to make the trigger more invisible and versatile, instead of keeping $\boldsymbol{\delta}$ universal, we optimize $g$ to be a trigger distribution $\mathcal{N}(\boldsymbol{\delta}, \boldsymbol{I})$ so that any noise draw from the trigger distribution will make the diffusion model generate target image. Formally, we make $g(\boldsymbol{\epsilon}) = \boldsymbol{\delta}'$ and $\mathcal{A}(P, g(P)) = \boldsymbol{\delta}' + \boldsymbol{\epsilon}, \boldsymbol{\delta}' \sim \mathcal{N}(\boldsymbol{\delta}, \boldsymbol{I}), \boldsymbol{\epsilon} \sim \mathcal{N}(\mathbf{0}, \boldsymbol{I})$. Keeping the trigger invisible, our trigger generator is dynamic and sample-specific, which is able to bypass the current universal perturbation-based detection and defense.

Given the above parameterization of $g$ and $\mathcal{A}$, we define the loss function $L_{outer}$ based on DDIM (Song et al., 2020) where we aim to create a secrect mapping for poisoned data between target image $y$ and poisoned distribution $\mathcal{N}(\boldsymbol{\delta}, \boldsymbol{I})$ or $\mathcal{N}(\boldsymbol{\delta}', \boldsymbol{I})$.

Let $\boldsymbol{x}_0' = \boldsymbol{y}$ (the target image distribution). $\boldsymbol{x}_1', \boldsymbol{x}_2', \cdots, \boldsymbol{x}_T' \in \mathbb{R}^d$ is then generated by gradually adding noise into $\boldsymbol{x}_0'$ where noise schedule is also controlled by $\beta_t$. The backdoored forward process is defined as:

$$q_\sigma(\boldsymbol{x}_{1:T}'|\boldsymbol{x}_0') := q_\sigma(\boldsymbol{x}_T'|\boldsymbol{x}_0') \prod_{t=2}^{T} q_\sigma(\boldsymbol{x}_{t-1}'|\boldsymbol{x}_t', \boldsymbol{x}_0'), \tag{9}$$

where $q_\sigma(\boldsymbol{x}_T'|\boldsymbol{x}_0') = \mathcal{N}(\sqrt{\bar{\alpha}_T}\boldsymbol{x}_0', (1 - \bar{\alpha}_T)\boldsymbol{I})$, and $q_\sigma(\boldsymbol{x}_{t-1}'|\boldsymbol{x}_t', \boldsymbol{x}_0') = \mathcal{N}(\sqrt{\bar{\alpha}_{t-1}}\boldsymbol{x}_0' + (1 - \sqrt{\bar{\alpha}_{t-1}})\boldsymbol{\delta} + \sqrt{1 - \bar{\alpha}_{t-1} - \sigma_t^2} \cdot \frac{\boldsymbol{x}_t' - \sqrt{\bar{\alpha}_t}\boldsymbol{x}_0' - (1 - \sqrt{\bar{\alpha}_t})\boldsymbol{\delta}}{\sqrt{1 - \bar{\alpha}_t}}, \sigma_t^2 \boldsymbol{I})$.

The mean function in $q_\sigma(\boldsymbol{x}_{t-1}'|\boldsymbol{x}_t', \boldsymbol{x}_0')$ is chosen to ensure that $q_\sigma(\boldsymbol{x}_t'|\boldsymbol{x}_0') := \mathcal{N}(\boldsymbol{x}_t'; \sqrt{\bar{\alpha}_t}\boldsymbol{x}_0' + (1 - \sqrt{\bar{\alpha}_t})\boldsymbol{\delta}, (1 - \bar{\alpha}_t)\boldsymbol{I})$ (See Appendix A for proof). Using DDIM sampling process, we can directly set $\sigma_t = 0$ to further simplify the derivation. the loss function based on the minimization of KL divergence between parameterized $p_\theta(\boldsymbol{x}_{t-1}'|\boldsymbol{x}_t')$ and $q_\sigma(\boldsymbol{x}_{t-1}'|\boldsymbol{x}_t', \boldsymbol{x}_0')$ can be written as

$$\mathbb{E}_{\boldsymbol{x}_0', \boldsymbol{\epsilon}, t} \left[ \left\| \boldsymbol{\epsilon} + \frac{\sqrt{\bar{\alpha}_{t-1}} - \sqrt{\bar{\alpha}_t}}{\sqrt{\bar{\alpha}_{t-1}}\sqrt{1 - \bar{\alpha}_t} - \sqrt{\bar{\alpha}_t}\sqrt{1 - \bar{\alpha}_{t-1}}} \boldsymbol{\delta} - \boldsymbol{\epsilon}_\theta(\boldsymbol{x}_t'(\boldsymbol{x}_0', \boldsymbol{\delta}, \boldsymbol{\epsilon}), t) \right\|^2 \right], \tag{10}$$

where $\boldsymbol{x}_t'(\boldsymbol{x}_0', \boldsymbol{\delta}, \boldsymbol{\epsilon}) = \sqrt{\bar{\alpha}_t}\boldsymbol{x}_0' + (1 - \sqrt{\bar{\alpha}_t})\boldsymbol{\delta} + \sqrt{1 - \bar{\alpha}_t}\boldsymbol{\epsilon}$. We defer the detailed derivation in the appendix A. To let the diffusion model learn different distribution mapping, we construct the poisoned dataset as $\mathcal{D} = \{\mathcal{D}_c, \mathcal{D}_p\}$ where $\mathcal{D}_c$ denotes the clean data and $\mathcal{D}_p$ is the poisoned data. Now we can combine the loss function for backdooring diffusion process with loss function for clean diffusion process to obtain $L_{outer}$ for the outer optimization. The training algorithm under the bi-level optimization framework is shown in Algorithm 1. During training, for poisoned sample and clean sample, we design the following loss function:

$$L_{outer}(\boldsymbol{\epsilon}_\theta, \boldsymbol{x}_0, \boldsymbol{\epsilon}, \boldsymbol{\delta}, \boldsymbol{y}, t) =$$
$$\begin{cases} \|\boldsymbol{\epsilon} - \boldsymbol{\epsilon}_\theta(\sqrt{\bar{\alpha}_t}\boldsymbol{x}_0 + \sqrt{1 - \bar{\alpha}_t}\boldsymbol{\epsilon}, t)\|^2, & \text{if } \boldsymbol{x}_0 \in \mathcal{D}_c, \\ \|\boldsymbol{\epsilon} + \frac{\sqrt{\bar{\alpha}_{t-1}} - \sqrt{\bar{\alpha}_t}}{\sqrt{\bar{\alpha}_{t-1}}\sqrt{1 - \bar{\alpha}_t} - \sqrt{\bar{\alpha}_t}\sqrt{1 - \bar{\alpha}_{t-1}}}\boldsymbol{\delta} - \boldsymbol{\epsilon}_\theta(\boldsymbol{x}_t'(\boldsymbol{y}, \boldsymbol{\delta}, \boldsymbol{\epsilon}), t)\|^2, & \text{if } \boldsymbol{x}_0 \in \mathcal{D}_p. \end{cases} \tag{11}$$

To be noted, although our loss is derived from DDIM, our proposed framework could choose a wide range of samplers such as DDIM (Song et al., 2020), DPMSolver (Lu et al., 2022) etc.

### 3.3.2 BACKDOORING CONDITIONAL DIFFUSION MODEL

Different from the above unconditional diffusion models in which the prior is random noise, conditional diffusion models can have various priors, such as texts, images, masked images, and even sketches (Ramesh et al., 2021; Nichol et al., 2021; Meng et al., 2021; Saharia et al., 2022; Rombach et al., 2022). Since the prior are all natural images and texts, it is thus crucial to make the trigger invisible where previously used visible triggers (Chou et al., 2023a; Chen et al., 2023b; Chou et al., 2023b) can be detected without any effort. For simplicity, we formulate how to learn invisible triggers for conditional diffusion model in the text-guided image editing pipeline used in (Nichol et al., 2021). For text-guided image editing, the priors consist of the masked image to be edited, a mask marking editing regions, and text (Nichol et al., 2021; Rombach et al., 2022). Unlike previous

---

**Algorithm 1** Backdoored diffusion model training given the prior is random noise, i.e., unconditional generation.

---

1: **Input:** $K, D$, stepsizes $\alpha$ and $\beta$, initializations $\boldsymbol{\delta}_0$ and $\theta_0$, target image $\boldsymbol{y}$, dataset $\mathcal{D} = \{\mathcal{D}_c, \mathcal{D}_p\}$.
2: **for** $k = 0, 1, 2, ..., K$ **do**
3:      Set $\boldsymbol{\delta}_k^0 = \boldsymbol{\delta}_{k-1}^D$ if $k > 0$ and $\boldsymbol{\delta}_0$ otherwise
4:      **for** $i = 1, ...., D$ **do**
5:          $\tilde{\boldsymbol{\epsilon}} \sim \mathcal{N}(\boldsymbol{0}, \boldsymbol{I})$
6:          $\tilde{\boldsymbol{\delta}}_k^i = \boldsymbol{\delta}_k^{i-1} - \alpha \nabla_{\boldsymbol{\delta}} L_{inner}(\boldsymbol{\epsilon}_{\theta_k}, \tilde{\boldsymbol{\epsilon}}, \boldsymbol{\delta}_k^{i-1})$
7:          $\boldsymbol{\delta}_k^i = \text{Proj}_{\|\cdot\|_\infty \leq C}(\tilde{\boldsymbol{\delta}}_k^i)$
8:      **end for**
9:      $\boldsymbol{x}_0 \sim \{\mathcal{D}_c, \mathcal{D}_p\}$
10:     $t \sim \text{Uniform}(\{1, \ldots, T\})$
11:     $\boldsymbol{\epsilon} \sim \mathcal{N}(\boldsymbol{0}, \boldsymbol{I})$
12:     Compute gradient $\nabla_\theta L_{outer}(\boldsymbol{\epsilon}_{\theta_k}, \boldsymbol{x}_0, \boldsymbol{\epsilon}, \boldsymbol{\delta}_k^D, \boldsymbol{y}, t)$
13:     Update $\theta_{k+1} = \theta_k - \beta \nabla_\theta L_{outer}(\boldsymbol{\epsilon}_{\theta_k}, \boldsymbol{x}_0, \boldsymbol{\epsilon}, \boldsymbol{\delta}_k^D, \boldsymbol{y}, t)$
14: **end for**

---

works (Struppek et al., 2022; Chou et al., 2023b) that insert the backdoor into text representation, to best of our knowledge, not only are we the first to propose a general framework to learn invisible triggers but also the first to show how to backdoor text-guided image editing/inpainting pipeline.

Let the masked image be $\tilde{\boldsymbol{x}} = \boldsymbol{x}_0 \odot \boldsymbol{M}$ where $\boldsymbol{M}$ is the binary mask. Let $c$ be the text instruction for editing. Then the priors can be written as $P = \{\tilde{\boldsymbol{x}}, \boldsymbol{M}, c\}$. The aim of invisible triggers in this pipeline is to only insert imperceptible perturbation into masked natural images $\tilde{\boldsymbol{x}}$ to backdoor diffusion models. In this setting, we have to ensure that there is no perturbation/trigger in the masked region, or the inserted trigger can be immediately detected since the pixel values must be 0 in the masked region. To this end, we parameterize the trigger generator $g$ with a simple 6-layer neural network to learn input-aware triggers given masked image $\tilde{\boldsymbol{x}}$, mask $\boldsymbol{M}$, and target image $\boldsymbol{y}$. Let $\boldsymbol{\delta}_{\boldsymbol{x}_0}^M = g(\tilde{\boldsymbol{x}}, \boldsymbol{M}, \boldsymbol{y})$ be the generated input-aware triggers. To ensure there is no perturbation in the masked region, we directly constrain the generated trigger to have zero value on the masked region.

We consider two kinds of masks, rectangular masks and free-form masks proposed in (Yu et al., 2019), which can mimic the user-specified masks used in real-world applications. Note that due to the existence of additional priors, the sampling process is different now. With the additional priors, classifier-free guidance is used to generate images conditioned on additional priors (Ho & Salimans, 2022; Nichol et al., 2021). The predicted noise is computed as $(1 - \gamma)\boldsymbol{\epsilon}_\theta(\boldsymbol{x}_t, t, \tilde{\boldsymbol{x}}, \boldsymbol{M}, \emptyset) + \gamma\boldsymbol{\epsilon}_\theta(\boldsymbol{x}_t, t, \tilde{\boldsymbol{x}}, \boldsymbol{M}, c)$ where $\gamma$ is a hyperparameter that controls the strength of guidance (Ho & Salimans, 2022).

As defined in the threat model, the attacker aims to generate the target image $\boldsymbol{y}$ when the backdoor trigger is activated. This asks the conditional diffusion model should output the same target image regardless of any given text. In other words, the diffusion models should predict the same noise whenever there are triggers in the masked image. By setting the text to be a empty string, we mimic the aforementioned procedure as $\boldsymbol{\epsilon}_\theta(\boldsymbol{x}_t, t, \tilde{\boldsymbol{x}} + \boldsymbol{\delta}_{\boldsymbol{x}_0}^M, \boldsymbol{M}, c) = \boldsymbol{\epsilon}_\theta(\boldsymbol{x}_t, t, \tilde{\boldsymbol{x}} + \boldsymbol{\delta}_{\boldsymbol{x}_0}^M, \boldsymbol{M}, \emptyset)$. Therefore, in the inner optimization, we perform the sampling process independently with $c$ or $\emptyset$, similar to the above unconditional sampling instead of classifier-free guidance sampling, to generate target image.

We summarize the training algorithm to insert backdoor in conditional diffusion models in Algorithm 2. Given $\boldsymbol{x}_0 \sim \{\mathcal{D}_c, \mathcal{D}_p\}$, for the clean training (i.e., $\boldsymbol{x}_0 \in \mathcal{D}_c$), we add noise to $\boldsymbol{x}_0$ and optimize $\boldsymbol{\epsilon}_\theta$ which takes noisy $\boldsymbol{x}_0, \boldsymbol{x}_0 \odot \boldsymbol{M}, \boldsymbol{M}$ and $c$ as input to predict noise. For the backdoor training (i.e., $\boldsymbol{x}_0 \in \mathcal{D}_p$), we firstly sample clean data $\boldsymbol{x}_c \sim \mathcal{D}_c$, compute the corresponding masked version as $(\boldsymbol{x}_c \odot \boldsymbol{M})$, and generate the injected trigger $\delta_{\boldsymbol{x}_c}^M$ for the masked image. Then we add noise to target image $\boldsymbol{y}$, and optimize $\boldsymbol{\epsilon}_\theta$ which takes noisy $\boldsymbol{y}, (\boldsymbol{x}_c \odot \boldsymbol{M} + \delta_{\boldsymbol{x}_c}^M), \boldsymbol{M}$, and $c$ as input, to predict the noise. Hence $L_{outer}$ for the outer optimization can be written as

$$L_{outer}(\boldsymbol{\epsilon}_\theta, \boldsymbol{x}_0, \boldsymbol{M}, c, \boldsymbol{\delta}_{\boldsymbol{x}_c}^M, \boldsymbol{y}, t) =$$

$$\begin{cases} \|\boldsymbol{\epsilon} - \boldsymbol{\epsilon}_\theta(\sqrt{\bar{\alpha}_t}\boldsymbol{x}_0 + \sqrt{1 - \bar{\alpha}_t}\boldsymbol{\epsilon}, t, \boldsymbol{x}_0 \odot \boldsymbol{M}, \boldsymbol{M}, c)\|^2, \text{ if } \boldsymbol{x}_0 \in \mathcal{D}_c, \\ \|\boldsymbol{\epsilon} - \boldsymbol{\epsilon}_\theta(\sqrt{\bar{\alpha}_t}\boldsymbol{y} + \sqrt{1 - \bar{\alpha}_t}\boldsymbol{\epsilon}, t, \boldsymbol{x}_c \odot \boldsymbol{M} + \boldsymbol{\delta}_{\boldsymbol{x}_c}^M, \boldsymbol{M}, c)\|^2, \text{ if } \boldsymbol{x}_0 \in \mathcal{D}_p. \end{cases} \quad (12)$$

---

**Algorithm 2** Backdoored diffusion model training given the priors are masked image, mask, and text, i.e., conditional generation.

---

1: **Input:** $K, D$, stepsizes $\alpha$ and $\beta$, initializations $g_0$ and $\theta_0$, target image $\boldsymbol{y}$, dataset $\mathcal{D} = \{\mathcal{D}_c, \mathcal{D}_p\}$, function GenerateRandomMask() for generating random masks.
2: **for** $k = 0, 1, 2, ..., K$ **do**
3:      Set $g_k^0 = g_{k-1}^D$ if $k > 0$ and $g_0$ otherwise
4:      **for** $i = 1, ...., D$ **do**
5:          $\boldsymbol{M}' \leftarrow$ GenerateRandomMask()
6:          $(\boldsymbol{x}_0, c) \sim \{\mathcal{D}_c, \mathcal{D}_p\}$. Set $c = \emptyset$ with probability 50%.
7:          $\tilde{\boldsymbol{\delta}}_{\boldsymbol{x}_0,i}^M = g_k^{i-1}(\boldsymbol{x}_0 \odot \boldsymbol{M}', \boldsymbol{M}', \boldsymbol{y}) \odot \boldsymbol{M}'$
8:          $\boldsymbol{\delta}_{\boldsymbol{x}_0,i}^M = \mathrm{Proj}_{\|\cdot\|_\infty \leq C}(\tilde{\boldsymbol{\delta}}_{\boldsymbol{x}_0,i}^M)$
9:          $g_k^i = g_k^{i-1} - \alpha \nabla_g L_{inner}(\boldsymbol{\epsilon}_{\theta_k}, \boldsymbol{x}_0 \odot \boldsymbol{M}', \boldsymbol{M}', c, \boldsymbol{\delta}_{\boldsymbol{x}_0,i}^M)$
10:      **end for**
11:      $(\boldsymbol{x}_0, c) \sim \{\mathcal{D}_c, \mathcal{D}_p\}$. Set $c = \emptyset$ with probability 50%.
12:      $t \sim$ Uniform($\{1, \ldots, T\}$)
13:      $\boldsymbol{M} \leftarrow$ GenerateRandomMask()
14:      **if** $\boldsymbol{x}_0 \in \mathcal{D}_p$ **then**
15:          Sample $\boldsymbol{x}_c \sim \mathcal{D}_c$
16:          $\tilde{\boldsymbol{\delta}}_{\boldsymbol{x}_c}^M = g_k^D(\boldsymbol{x}_c \odot \boldsymbol{M}, \boldsymbol{M}, \boldsymbol{y}) \odot \boldsymbol{M}$
17:          $\boldsymbol{\delta}_{\boldsymbol{x}_c}^M = \mathrm{Proj}_{\|\cdot\|_\infty \leq C}(\tilde{\boldsymbol{\delta}}_{\boldsymbol{x}_c}^M)$
18:      **end if**
19:      Compute gradient $\nabla_\theta L_{outer}\left(\boldsymbol{\epsilon}_\theta, \boldsymbol{x}_0, \boldsymbol{x}_0 \odot \boldsymbol{M}, \boldsymbol{M}, c, \boldsymbol{\delta}_{\boldsymbol{x}_c}^M, \boldsymbol{y}, t\right)$
20:      Update $\theta_{k+1} = \theta_k - \beta \nabla_\theta L_{outer}\left(\boldsymbol{\epsilon}_\theta, \boldsymbol{x}_0, \boldsymbol{x}_0 \odot \boldsymbol{M}, \boldsymbol{M}, c, \boldsymbol{\delta}_{\boldsymbol{x}_c}^M, \boldsymbol{y}, t\right)$
21: **end for**

---

# 4 EXPERIMENTAL RESULTS

## 4.1 IMPLEMENTATION DETAILS

To verify the effectiveness and stealthiness of the proposed framework, we conduct extensive experiments in this section. For unconditional generation, we conduct the experiments on two commonly used datasets, CIFAR10($32 \times 32$) (Krizhevsky et al., 2009) and CELEBA-HQ($256 \times 256$) (Liu et al., 2015) used in Chou et al. (2023a;b). For conditional generation, we follow the text-guided image editing/inpainting pipeline in Nichol et al. (2021) and use the dataset MS COCO($64 \times 64$) (Lin et al., 2014). The diffusion models are trained from scratch for 400 epochs on both CIFAR10 and CELEBA-HQ for unconditional generation and we will also show that finetuning pre-trained models with less epochs is also feasible to inject the proposed invisible backdoor. For conditional case, we found that only finetuning for 5 epochs on about 10K images of MS COCO training data is enough to learn input-aware invisible backdoor. The learning rate of inner optimization is $1e-3$ for all cases, and for outer optimization, the learning rates are $2e-4$, $8e-5$, and $5e-4$ for CIFAR10, CELEBA-HQ, and MS COCO, respectively. To make inner optimization feasible, we sample 10, 3, and 5 steps to generate target images with DDIM (Song et al., 2020) sampling for CIFAR10, CELEBA-HQ, and MS COCO, respectively. All unconditional generation experiments are conducted on a single NVIDIA 3090 GPU, and all conditional generation experiments are conducted on a single NVIDIA A6000 GPU. Training on CIFAR10 and CELEBA-HQ from scratch spends about 3 days and 12 days respectively due to the large image resolution of CELEBA-HQ and multiple sampling steps in the inner optimization, which can be accelerated by more powerful GPUs. Training on MS COCO could be finished within about 30 mins since it only requires finetuning for 5 epochs. We mainly use three target images corresponding to the 'Hat', 'Shoe', and 'Cat' target used in Chou et al. (2023a;b). To evaluate the performance of backdoored model on utility and specificity, for unconditional case, we use FID (Heusel et al., 2017) to evaluate the utility, and MSE to evaluate the specificity. We sample about 50K and 10K images to compute FID and MSE for CIFAR10 and CELEBA-HQ, respectively. For conditional case, we use FID and LPIPS (Zhang et al., 2018) to evaluate the utility and MSE to evaluate the specificity. We sample the same number of images as the MS COCO validation set (about 40K) to compute FID, LPIPS, and MSE.

## 4.2 UNCONDITIONAL GENERATION RESULTS

**Universal backdoor triggers**    Firstly, we show the results for learning one universal invisible trigger on CIFAR10 and CELEBA-HQ. For CIFAR10, $\ell_\infty$ norm bound is set as 0.2 and the poison rate is 0.05. The 'Hat' image is the corresponding target image. As shown in Figure 2 and Table 1, the backdoored model can achieve similar FID with clean model and low MSE simultaneously while keeping the trigger invisible, demonstrating the high-utility and high-specificity as required by successful backdoor attack.

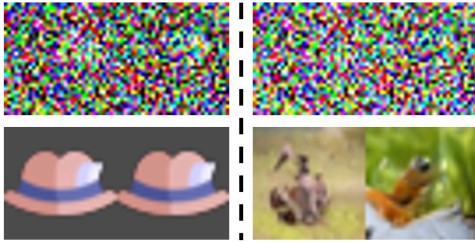

Figure 2: Visualization results for learnable universal trigger on CIFAR10.

Figure 3: Visualization results for learnable universal trigger on CELEBA-HQ.

|  | FID | MSE |
|---|---|---|
| Clean model | 12.80 | - |
| Backdoored model | 11.76 | 3.07e-3 |

Table 1: Quantitative results for learnable universal trigger on CIFAR10.

|  | FID | MSE |
|---|---|---|
| Clean model | 12.39 | - |
| Backdoored model | 11.19 | 4.57e-3 |

Table 2: Quantitative results for learnable universal trigger on CELEBA-HQ.

We show the results on high-resolution datasets CELEBA-HQ($256 \times 256$) in Figure 3 and Table 2, where the $\ell_\infty$ norm bound is also 0.2 and the poison rate is 0.3. The 'Cat' image is the corresponding target image. It could clearly observed that the proposed invisible trigger is still feasible and effective for high-resolution images. To further show the capability of the proposed framework, we show it is possible to learn multiple universal trigger-target pairs simultaneously. Examples are available in Appendix D.

In addition, we also conducted experiments on different samplers, DPMSolver (Lu et al., 2022) to show that the proposed loss in unconditional generation can be directly applied to different commonly used samplers. Detailed results are shown in Appendix E.

**Distribution based trigger results**    As stated in Section 3, to make the invisible trigger even more stealthy, we can optimize the trigger distribution instead of universal trigger so that we are able to generate sample-specific triggers. The results are shown in Figure 4 and Table 3. Compared with universal trigger, our distribution based trigger achieve a even smaller gap on the FID while keeping an excellent performance on generating target image with very small MSE.

|  | FID | MSE |
|---|---|---|
| Clean model | 12.80 | - |
| Backdoored model | 12.86 | 1.82e-5 |

Table 3: Quantitative results for learnable trigger distribution on CIFAR10.

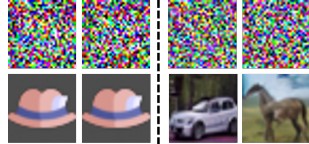

Figure 4: Visualization results for learnable trigger distribution on CIFAR10.

## 4.3 CONDITIONAL GENERATION RESULTS

In this section, we show the results in text-guided image editing/inpainting pipeline on MS COCO dataset (Lin et al., 2014). For simplicity, we ignore the text part in visualization results since the proposed framework will generate the target image given any text if the backdoor is triggered. By setting the norm bound as 0.04, we show the quantitative results on evaluation metrics and visualization results in Figure 5 and Table 4, where the target image is the 'Hat' image. The results

indicate that the backdoored model perform similarly to clean model when there are no triggers in the inputs, and generate target image when triggers are injected into inputs. Figure 6 and 7 shows

| | FID | LPIPS | MSE |
|---|---|---|---|
| Clean model | 1.00 | 0.064 | - |
| Backdoored model | 1.01 | 0.064 | 6.85e-3 |

Table 4: Quantitative results for learnable input-aware trigger in conditional diffusion models.

Figure 5: Visualization results for learnable input-aware trigger in conditional diffusion models.

the visualization results under different norm bounds. With larger norm bound, we can expect larger perturbations in the generated triggers, which is illustrated in the visualization results. We also

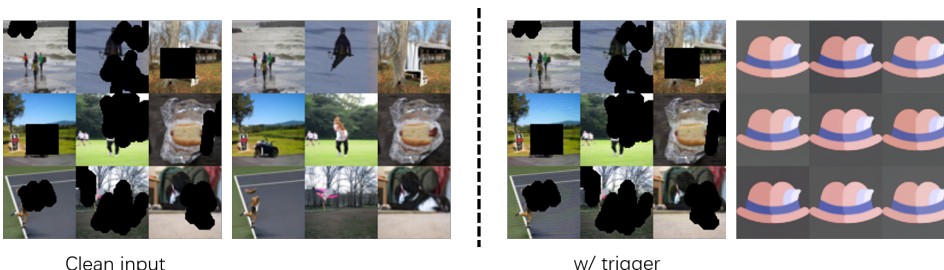

Clean input                w/ trigger

Figure 6: Visualization results with norm bound 0.04.

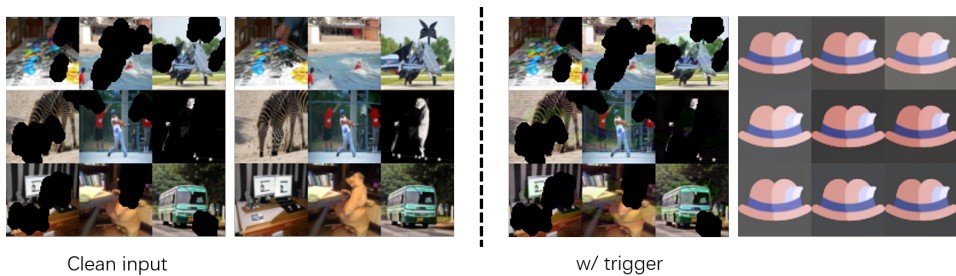

Clean input                w/ trigger

Figure 7: Visualization results with norm bound 0.06.

conduct experiments to show it is possible to insert multiple targets in this case. Please refer to Appendix F for details.

## 4.4 DEFENSE AGAINST BACKDOOR ATTACK IN DIFFUSION MODELS

The above results have shown the powerful capability of our proposed attack, which may lead to severe consequences in practice. Hence defense for mitigating backdoors is crucial. By effectively mitigating the backdoor vulnerabilities in diffusion models, we can ensure their safe utilization in real-world implementations for safety-critical applications. In addition, diffusion models without backdoors can be confidently employed by individuals for everyday image generation tasks.

Previous work (Chou et al., 2023a) has tested the effectiveness of two defense methods against backdoor in diffusion models, namely Adversarial neuron pruning (Wu & Wang, 2021) and inference-time clipping. They show the inference-time clipping, which clips the latent generation during sampling to the range $[-1, 1]$, is effective on their proposed attack. However, we found that both of

them become totally ineffective in our proposed framework, which indicates more advanced defense methods need to be developed for diffusion models. For detailed results, please refer to Appendix G.

### 4.5 ABLATION STUDY

To show the effect of poison rate and norm bound, we conduct ablation studies on unconditional generation case. Please refer to Appendix H for more results. We also show that finetuning can be easily done to inject backdoor under our framework. Because of space limit, we defer the results in Appendix I.

## 5 CONCLUSION

In this paper, we propose a novel and general optimization framework to learn input-aware invisible triggers to backdoor diffusion models, which can be used for both unconditional and conditional diffusion models. Also, to make the optimization in unconditional case feasible, we derive a loss function based on DDIM (Song et al., 2020) sampling. In addition, we are also the first to show how to backdoor text-guided image editing pipeline in conditional diffusion models. Our proposed framework reveals severe security threat that diffusion models may bring. For future work, we will explore effective defense methods to mitigate possible backdoor in diffusion models.

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
