## A  Loss function based on DDIM sampling

As we show in Equation 9, $q_\sigma(\boldsymbol{x}'_{t-1}|\boldsymbol{x}'_t, \boldsymbol{x}'_0) = \mathcal{N}(\sqrt{\bar{\alpha}_{t-1}}\boldsymbol{x}'_0 + (1 - \sqrt{\bar{\alpha}_{t-1}})\boldsymbol{\delta} + \sqrt{1 - \bar{\alpha}_{t-1} - \sigma_t^2} \cdot$
$\frac{\boldsymbol{x}'_t - \sqrt{\bar{\alpha}_t}\boldsymbol{x}'_0 - (1 - \sqrt{\bar{\alpha}_t})\boldsymbol{\delta}}{\sqrt{1 - \bar{\alpha}_t}}, \sigma_t^2\boldsymbol{I})$, where the mean function is chosen to ensure that $q_\sigma(\boldsymbol{x}'_t|\boldsymbol{x}'_0) =$
$\mathcal{N}(\boldsymbol{x}'_t; \sqrt{\bar{\alpha}_t}\boldsymbol{x}'_0 + (1 - \sqrt{\bar{\alpha}_t})\boldsymbol{\delta}, (1 - \bar{\alpha}_t)\boldsymbol{I})$. We provide the proof in the following.

**Lemma 1.** *Let $q_\sigma(\boldsymbol{x}'_{1:T}|\boldsymbol{x}'_0)$ and $q_\sigma(\boldsymbol{x}'_{t-1}|\boldsymbol{x}'_t, \boldsymbol{x}'_0)$ be defined by Equation 9, we have*

$$q_\sigma(\boldsymbol{x}'_t|\boldsymbol{x}'_0) = \mathcal{N}(\boldsymbol{x}'_t; \sqrt{\bar{\alpha}_t}\boldsymbol{x}'_0 + (1 - \sqrt{\bar{\alpha}_t})\boldsymbol{\delta}, (1 - \bar{\alpha}_t)\boldsymbol{I}).$$

*Proof.* To prove it, we use a similar way to Song et al. (2020). Assume $\forall t \leq T$, $q_\sigma(\boldsymbol{x}'_t|\boldsymbol{x}'_0) = \mathcal{N}(\sqrt{\bar{\alpha}_t}\boldsymbol{x}'_0 + (1 - \sqrt{\bar{\alpha}_t})\boldsymbol{\delta}, (1 - \bar{\alpha}_t)\boldsymbol{I})$. Now if we can prove $q_\sigma(\boldsymbol{x}'_{t-1}|\boldsymbol{x}'_0) = \mathcal{N}(\sqrt{\bar{\alpha}_{t-1}}\boldsymbol{x}'_0 + (1 - \sqrt{\bar{\alpha}_{t-1}})\boldsymbol{\delta}, (1 - \bar{\alpha}_{t-1})\boldsymbol{I})$, then the statement can be proved by induction.

We have

$$q_\sigma(\boldsymbol{x}'_{t-1}|\boldsymbol{x}'_0) = \int_{\boldsymbol{x}'_t} q_\sigma(\boldsymbol{x}'_t|\boldsymbol{x}'_0) q_\sigma(\boldsymbol{x}'_{t-1}|\boldsymbol{x}'_t, \boldsymbol{x}'_0) d\boldsymbol{x}'_t, \tag{13}$$

where

$$q_\sigma(\boldsymbol{x}'_t|\boldsymbol{x}'_0) = \mathcal{N}(\sqrt{\bar{\alpha}_t}\boldsymbol{x}'_0 + (1 - \sqrt{\bar{\alpha}_t})\boldsymbol{\delta}, (1 - \bar{\alpha}_t)\boldsymbol{I}),$$

$$q_\sigma(\boldsymbol{x}'_{t-1}|\boldsymbol{x}'_t, \boldsymbol{x}'_0) = \mathcal{N}(\sqrt{\bar{\alpha}_{t-1}}\boldsymbol{x}'_0 + (1 - \sqrt{\bar{\alpha}_{t-1}})\boldsymbol{\delta} + \sqrt{1 - \bar{\alpha}_{t-1} - \sigma_t^2} \cdot \frac{\boldsymbol{x}'_t - \sqrt{\bar{\alpha}_t}\boldsymbol{x}'_0 - (1 - \sqrt{\bar{\alpha}_t})\boldsymbol{\delta}}{\sqrt{1 - \bar{\alpha}_t}}, \sigma_t^2\boldsymbol{I}).$$

Then from Bishop & Nasrabadi (2006) (2.115), we can write $q_\sigma(\boldsymbol{x}'_{t-1}|\boldsymbol{x}'_0)$ as Gaussian distribution, where the mean

$$\boldsymbol{\mu} = \sqrt{\bar{\alpha}_{t-1}}\boldsymbol{x}'_0 + (1 - \sqrt{\bar{\alpha}_{t-1}}\boldsymbol{\delta} + \sqrt{1 - \bar{\alpha}_{t-1} - \sigma_t^2} \cdot \frac{\sqrt{\bar{\alpha}_t}\boldsymbol{x}'_0 + (1 - \sqrt{\bar{\alpha}_t})\boldsymbol{\delta} - \sqrt{\bar{\alpha}_t}\boldsymbol{x}'_0 - (1 - \sqrt{\bar{\alpha}_t})\boldsymbol{\delta}}{\sqrt{1 - \bar{\alpha}_t}}$$

$$= \sqrt{\bar{\alpha}_{t-1}}\boldsymbol{x}'_0 + (1 - \sqrt{\bar{\alpha}_{t-1}})\boldsymbol{\delta}, \tag{14}$$

variance

$$\boldsymbol{\Sigma} = \left(\frac{1 - \bar{\alpha}_{t-1} - \sigma_t^2}{1 - \bar{\alpha}_t} \cdot (1 - \bar{\alpha}_t)\right)\boldsymbol{I} + \sigma_t^2\boldsymbol{I} = (1 - \bar{\alpha}_{t-1})\boldsymbol{I}. \tag{15}$$

Hence

$$q_\sigma(\boldsymbol{x}'_{t-1}|\boldsymbol{x}'_0) = \mathcal{N}(\sqrt{\bar{\alpha}_{t-1}}\boldsymbol{x}'_0 + (1 - \sqrt{\bar{\alpha}_{t-1}})\boldsymbol{\delta}, (1 - \bar{\alpha}_{t-1})\boldsymbol{I}), \tag{16}$$

which finishes the proof. $\square$

Since we consider DDIM sampling, we can set $\sigma_t = 0$ to simplify the derivation. On the other hand, from $q_\sigma(\boldsymbol{x}'_t|\boldsymbol{x}'_0) = \mathcal{N}(\boldsymbol{x}'_t; \sqrt{\bar{\alpha}_t}\boldsymbol{x}'_0 + (1 - \sqrt{\bar{\alpha}_t})\boldsymbol{\delta}, (1 - \bar{\alpha}_t)\boldsymbol{I})$, we have

$$\boldsymbol{x}'_0 = \frac{\boldsymbol{x}'_t - \sqrt{1 - \bar{\alpha}_t}\boldsymbol{\epsilon} - (1 - \sqrt{\bar{\alpha}_t})\boldsymbol{\delta}}{\sqrt{\bar{\alpha}_t}}, \boldsymbol{\epsilon} \sim \mathcal{N}(\boldsymbol{0}, \boldsymbol{I}). \tag{17}$$

Given the above reverse transition $q_\sigma(\boldsymbol{x}'_{t-1}|\boldsymbol{x}'_t, \boldsymbol{x}'_0) = \mathcal{N}(\sqrt{\bar{\alpha}_{t-1}}\boldsymbol{x}'_0 + (1 - \sqrt{\bar{\alpha}_{t-1}})\boldsymbol{\delta} + \sqrt{1 - \bar{\alpha}_{t-1} - \sigma_t^2} \cdot \frac{\boldsymbol{x}'_t - \sqrt{\bar{\alpha}_t}\boldsymbol{x}'_0 - (1 - \sqrt{\bar{\alpha}_t})\boldsymbol{\delta}}{\sqrt{1 - \bar{\alpha}_t}}, \sigma_t^2\boldsymbol{I})$, substitute $\boldsymbol{x}'_0$ with Equation 17. After rearranging the common terms, the reverse transition can be rewritten as

$$q_\sigma(\boldsymbol{x}'_{t-1}|\boldsymbol{x}'_t, \boldsymbol{x}'_0) =$$
$$\frac{\sqrt{\bar{\alpha}_{t-1}}}{\sqrt{\bar{\alpha}_t}}\left[\boldsymbol{x}'_t - \frac{\sqrt{\bar{\alpha}_{t-1}}\sqrt{1 - \bar{\alpha}_t} - \sqrt{\bar{\alpha}_t}\sqrt{1 - \bar{\alpha}_{t-1}}}{\sqrt{\bar{\alpha}_{t-1}}}\left(\boldsymbol{\epsilon} + \frac{\sqrt{\bar{\alpha}_{t-1}} - \sqrt{\bar{\alpha}_t}}{\sqrt{\bar{\alpha}_{t-1}}\sqrt{1 - \bar{\alpha}_t} - \sqrt{\bar{\alpha}_t}\sqrt{1 - \bar{\alpha}_{t-1}}}\boldsymbol{\delta}\right)\right]. \tag{18}$$

Equation 18 indicates that now we need to train the network to predict $\boldsymbol{\epsilon} + \frac{\sqrt{\bar{\alpha}_{t-1}} - \sqrt{\bar{\alpha}_t}}{\sqrt{\bar{\alpha}_{t-1}}\sqrt{1 - \bar{\alpha}_t} - \sqrt{\bar{\alpha}_t}\sqrt{1 - \bar{\alpha}_{t-1}}}\boldsymbol{\delta}$ instead of only $\boldsymbol{\epsilon}$ for backdoor training, which leads to the loss function in Equation 10.

## B    DISCUSSION ON THE IMPORTANCE AND MOTIVATION OF BACKDOORING DIFFUSION MODELS WITH INVISIBLE TRIGGERS

As also discussed in previous work (Chou et al., 2023a;b; Chen et al., 2023b), backdooring diffusion models is an important topic for safe utilization of diffusion models. Since the powerful models like Stable Diffusion (Rombach et al., 2022) is open-sourced, anyone could download the model and conduct malicious fine-tuning to insert a secret backdoor that can exhibit a designated action (e.g. generating a inappropriate or incorrect images). Explicitly, the generated output will be directly controlled by activating backdoor for conducting some bad actions like disseminating propaganda, generating fake contents etc. Meanwhile, implicitly, as also discussed in (Chou et al., 2023a), the diffusion model has been widely used in a lot of different downstream tasks and applications such as reinforcement learning, object detection, and semantic segmentation (Baranchuk et al., 2021; Chen et al., 2022; 2023a). Hence if the diffusion model is backdoored, this Trojan effect can bring immeasurable cartographic damage to all downstream tasks and applications.

Given the importance of backdooring diffusion models, exploring invisibility of image triggers could further help the community understand the potential security threat better. As both mentioned in (Doan et al., 2021b;a), it is important to improve the fidelity of poisoned examples that are used to inject the backdoor and hence reduce the perceptual detectability by human observers. In the unconditional case, it is thus important to make the sampled noise to be similar with random noise used in the practice or it could be easily filtered by human inspection. As shown in Figure 1 and Figure 2, the triggers used by previous works (also in the unconditional case) could be easily detected through human inspection without any effort. In contrast, our proposed invisible trigger is nearly visually indistinguishable from the original input, which greatly increase attack's stealth so that human inspection would no longer effective. In addition to unconditional generation, invisible triggers are particularly practical in conditional diffusion models, which hasn't been explored and discussed by the previous works.

## C    DISCUSSION ON LEARNABLE INVISIBLE TRIGGERS THROUGH BI-LEVEL OPTIMIZATION IN CLASSIFICATION MODELS

As mentioned in Section 1, learning invisible triggers by bi-level optimization in diffusion models is different and much harder compared to finding one in classification models. The method developed for backdooring classification models cannot be directly or easily extended to backdoor diffusion models. Specifically, the threat model is totally different. Diffusion models consist of diffusion and reverse processes that fundamentally differs from classification models. Backdooring diffusion model needs to have careful control of the training procedure while only poisoning data needs to be added in the classification model. At the same time, it is nontrivial and challenging to design the backdoor objective in the conditional and unconditional diffusion model while it is relatively a simple task in the classification. To learn invisible backdoors for both unconditional and conditional diffusion models, the entire pipeline, training paradigm, and training loss have to be redesigned to differ significantly when applying bi-level optimization to backdoor diffusion models. In this setting, the training loss, training paradigm, and pipeline are specifically designed based on the properties of diffusion models differing substantially from backdooring classification models through bi-level optimization.

## D    MULTIPLE UNIVERSAL TRIGGER-TARGET PAIRS

To further show the capability of the proposed framework, we show it is possible to learn multiple universal trigger-target pairs simultaneously. The results with two trigger-target pairs on CIFAR10 are shown in Table 5, which indicate that the framework can be directly extended to learn multiple universal trigger-target pairs.

## E    EXPERIMENTS ON DIFFERENT SAMPLERS

We also conducted experiments on different samplers, DPMSolver (Lu et al., 2022) to show that the proposed loss in unconditional generation can be directly applied to different commonly used

| | FID | MSE for first target | MSE for second target |
|---|---|---|---|
| Clean model | 12.80 | - | - |
| Backdoored model | 13.77 | 4.40e-3 | 2.33e-6 |

Table 5: Quantitative results on two universal trigger-target pairs.

samplers. Previous work (Chou et al., 2023a) only consider DDPM sampling and the trained backdoored diffusion models cannot be used for other samplers. Figure 8 shows the sampling results with previous work's backdoor models where the left one is triggered inputs and the right one is sampling results, indicating the backdoor is ineffective when other samplers are used. In our proposed framework, however, different commonly used samplers can be used. We use second-order DPMSolver to test the backdoor performance. As shown in Figure 9 and Table 6, the injected backdoor is still very effective.

| | FID | MSE |
|---|---|---|
| Clean model | 12.80 | - |
| Backdoored model | 9.50 | 3.10e-3 |

Table 6: Quantitative results for DPMSolver sampler.

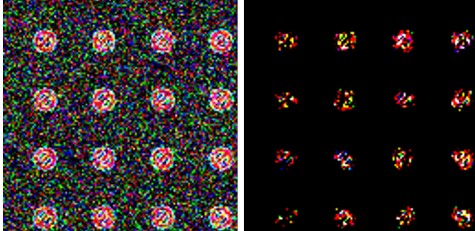

Figure 8: Illustration of previous backdoor (Chou et al., 2023a) for DDIM sampler on CIFAR10. The left figure is the initial noise with the visible triggered and the right figure is the generated output from sampling.

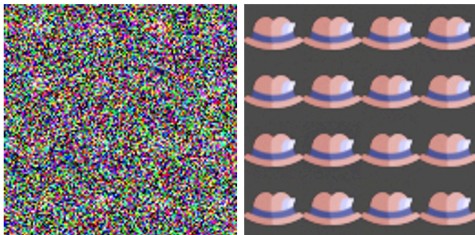

Figure 9: Visualization results for DPM-Solver sampler on CIFAR10.

## F    MULTIPLE INPUT-AWARE TRIGGER-TARGET PAIRS

Recall that for the conditional case, the inputs to the trigger generator are masked image, mask, and target image. Hence if we use different target images, can we insert multiple targets simultaneously? Here we show it is possible to insert multiple targets during training. Specifically, we use two target images ('Hat' and 'Shoe') to train the trigger generator to learn input-aware invisible triggers based on masked image and target image. The results are shown in Table 7.

## G    DEFENSE AGAINST BACKDOORED DIFFUSION MODELS

In this section, we show that both ANP (Wu & Wang, 2021) and inference-time clipping become totally ineffective in our proposed framework. We first present defense results on ANP against a backdoored diffusion model trained on CIFAR10 with norm bound 0.2 and poison rate 0.1. Following the settings in (Chou et al., 2023a), we use the largest perturbation budget (budget=4, larger budge means better Trojan detection) in (Chou et al., 2023a) and train the perturbated model with the whole clean dataset for 5 epochs. With different learning rates($1e − 4, 2e − 4$), we found ANP performs even worse on our proposed attack, compared to the performance on the attack in (Chou et al., 2023a). The perturbated model immediately collapses to a meaningless image or a black image. The visualization results with different learning rates during the training are shown in Figure 10 and Figure 11. This can also observed from the MSE results between the reversed target image by ANP and the true target image ('Hat' in our experiments), as shown in Table 8 and Table 9. We

|  | FID | LPIPS | MSE for first target | MSE for second target |
|---|---|---|---|---|
| Clean model | 1.00 | 0.064 | - | - |
| Backdoored model | 1.02 | 0.063 | 1.44e-3 | 2.07e-3 |

Table 7: Quantitative results on two target images.

sample 2048 images to compute the MSE, same as (Chou et al., 2023a). As shown in the tables, the computed MSE values are large, indicating that ANP cannot reconstruct the target image at all.

Secondly, we demonstrate the defense results of inference-time clipping. With the clip operation in Chou et al. (2023a), we sample images with DDIM (Song et al., 2020) sampling with different poison rates. As shown in Figure 12 and Table 10, with clipping, backdoored models can still achieve high-utility and high-specificity, which indicates the defense method is not a good choice for these cases.



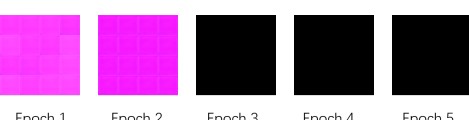

Figure 10: Reversed target images by ANP with learning rate $1e - 4$.

Figure 11: Reversed target images by ANP with learning rate $2e - 4$.

| Epoch | 1 | 2 | 3 | 4 | 5 |
|---|---|---|---|---|---|
| MSE | 0.28 | 0.28 | 0.22 | 0.19 | 0.24 |

Table 8: MSE between reversed target images and the true target image. Learning rate: $1e - 4$.

| Epoch | 1 | 2 | 3 | 4 | 5 |
|---|---|---|---|---|---|
| MSE | 0.24 | 0.26 | 0.24 | 0.24 | 0.24 |

Table 9: MSE between reversed target images and the true target image. Learning rate: $2e - 4$.

## H   ABLATION STUDY

Recall from our optimization framework that we projected generated triggers into an $\ell_\infty$ norm ball for trigger invisibility. Here, we investigate how different norm values may influence our model. Figure 13 shows the visualization results of influence of different values of norm. With larger norm bound, perturbations also become larger, which is what we can expect.

Quantitative results on two different targets ('Hat' and 'Cat') are shown in Table 11. The FID corresponding to clean model is 12.80, as shown in Table 1.

We have found that with our optimization framework, even with low norm value of 0.1 (invisibility), we can still successfully implant backdoor (specificity) while achieving the comparable FID score(utility) when compared to the clean model. This result provides insights when applying the proposed framework to conditional settings where trigger invisibility is crucial for the stealthy nature of our implanted backdoor.

We also performed experiments to demonstrate the impact of different poison rates, as illustrated in Figure 14 and 15. As the poison rate increases, we observe that the FID score increases while the MSE (Mean Squared Error) decreases. This aligns with our expectations since a larger poison rate implies a more substantial impact on clean performance. When employing a high poison rate (e.g., poison rate of 0.5), we find that the FID remains comparable to that of the clean model. This observation suggests that the proposed framework maintains its effectiveness across a range of settings and is resilient even under substantial poisoning conditions.

## I   RESULTS ON FINETUNING PRETRAINED MODELS

Here, we showcase the effectiveness of our proposed framework by fine-tuning pre-trained models for varying numbers of epochs. The results are presented in Table 12. It is worth noting that we can

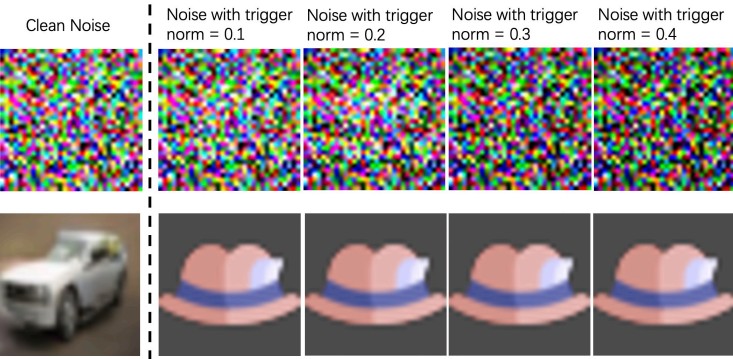

Figure 13: Visualization results for different norm bounds.

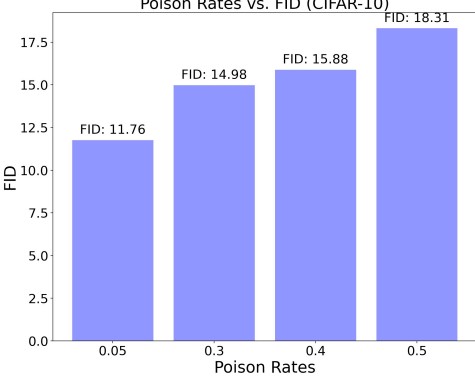

Figure 14: FID results for different poison rates on CIFAR10.

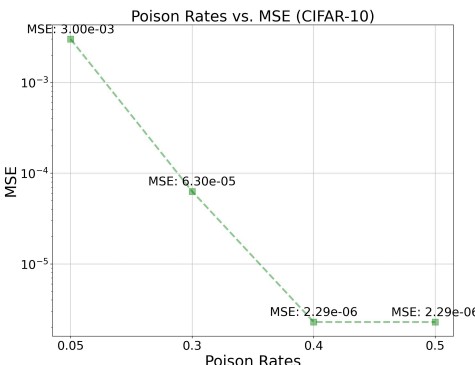

Figure 15: MSE results for different poison rates on CIFAR10.

| Poison rate | w/o clip | | w/ clip | |
|---|---|---|---|---|
| | FID | MSE | FID | MSE |
| 0.05 | 11.76 | 3.07e-3 | 11.76 | 3.49e-3 |
| 0.3 | 14.98 | 6.36e-5 | 14.98 | 8.59e-5 |
| 0.4 | 15.88 | 2.29e-6 | 15.88 | 2.29e-6 |
| 0.5 | 18.31 | 2.29e-6 | 18.33 | 2.29e-6 |

Table 10: Quantitative results w/ and w/o clip operation on CIFAR10.

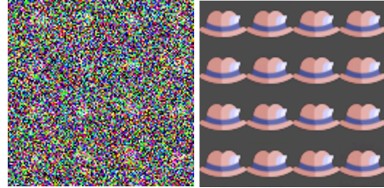

Figure 12: Visualization results w/ clip operation for backdoor sampling.

| Norm | Target 'Hat' | | Target 'Cat' | |
|---|---|---|---|---|
| | FID | MSE | FID | MSE |
| 0.1 | 13.01 | 2.28e-3 | 12.92 | 2.75e-3 |
| 0.2 | 12.44 | 8.13e-5 | 12.56 | 1.01e-5 |
| 0.3 | 12.38 | 1.06e-3 | 12.69 | 6.00e-4 |
| 0.4 | 12.35 | 1.92e-4 | 12.93 | 2.77e-6 |

Table 11: Results on different norm bounds on CIFAR10.

successfully introduce a backdoor into the model by fine-tuning it for as few as 30 epochs, yet still achieve a lower FID compared to the clean model. This means the proposed framework can easily be applied in practice.

| Finetuning epochs | Poison rate=0.1 | | Poison rate=0.5 | |
|---|---|---|---|---|
| | FID | MSE | FID | MSE |
| 30 | 8.22 | 3e-5 | 8.55 | 3.14e-6 |
| 100 | 6.40 | 6.12e-6 | 6.20 | 2.34e-6 |

Table 12: Results on finetuning pre-trained models with different poison rates.