# OpenReview forum: "Learnable Invisible Backdoor for Diffusion Models"
_ICLR.cc/2024/Conference — Submitted to ICLR 2024_

### Official Review · Reviewer_RZ4Z · 2023-10-29

**Soundness:** 3 good
**Presentation:** 3 good
**Contribution:** 3 good
**Rating:** 8
**Confidence:** 3

**Summary:**

The paper delves into the realm of diffusion models, which are vital for high-quality image generation but come with potential security risks. A significant challenge has been the vulnerability of these models to backdoor attacks, primarily using noticeable, manually-designed triggers. The authors introduce a cutting-edge optimization framework that develops undetectable triggers, enhancing the stealth and robustness of the inserted backdoor. This framework is adaptable to both unconditional and conditional diffusion models, with the latter's application to text-guided image editing being a pioneering contribution. Conclusively, this research not only establishes the profound security vulnerabilities these models may possess but also hints at the prospective exploration of effective defense mechanisms. Based on the insights provided, my recommendation is to accept the paper with minor modifications.

**Strengths:**

•	Innovation & Novelty: The paper opens doors to a fresh domain by presenting an innovative approach towards generating invisible triggers, setting it apart from conventional methods. The introduction of backdoor capabilities for text-guided image editing in conditional diffusion models further signifies its novelty.
•	Experimentation & Analysis: The authors have conducted extensive experiments on various samplers and datasets, solidifying the efficacy of the proposed model. Such detailed empirical evidence enhances the paper's credibility.
•	Clarity & Supplementary Materials: The research is presented with clarity, making it comprehensible. The inclusion of supplementary materials augments the paper's integrity.

**Weaknesses:**

•	Lack of Training Details: The paper omits crucial details related to the overall training time and specifics about hyper-parameter tuning. This information is vital for replication and a deeper understanding of the proposed framework.
•	Risk Mitigation & Practical Implementation: While the paper acknowledges the security risks, it could benefit from a more comprehensive discussion on the practical implications of mitigating these risks and the challenges in real-world implementation.

**Questions:**

•	Convergence Concerns: The paper doesn't fully address the convergence issues, especially considering the two-level optimization problem. There is ambiguity on how to ensure convergence and the steps to be taken if inner and/or outer problem doesn't converge.

---

> ### Author Response · Authors · 2023-11-22
> **Response to Reviewer RZ4Z**
>
> Thanks a lot for the suggestions. We have addressed your concerns in the following.
>
> **Q: Training details**
>
> A: Thanks for the advice. We have incorporated additional training details in our revised manuscript to provide a more comprehensive understanding of our methodology. Please refer to Section 4.1(marked in blue) in the revised manuscript. Furthermore, to promote further research and collaboration, we make our code publicly available on https://github.com/invisibleTriggerDiffusion/invisible_triggers_for_diffusion.
>
> **Q: Practical implications of mitigating the risks**
>
> A: Thanks for the advice. We aim to delve further into the practical implications of mitigating these risks and have integrated this discussion into our revised manuscript. Please refer to Section 4.4(marked in blue) for details. As mentioned in our manuscript and previous research on backdooring diffusion models [a], a backdoored diffusion model has the potential to generate images containing violent or erotic elements, leading to a highly biased dataset when utilized in practical applications. Moreover, the application of a backdoored diffusion model in downstream tasks, such as reinforcement learning, object detection, semantic segmentation [b, c, d], etc., can have severe consequences.
>
> By effectively mitigating the backdoor vulnerabilities in diffusion models, we can ensure their safe utilization in real-world implementations for safety-critical applications. In addition, diffusion models without backdoors can be confidently employed by individuals for everyday image generation tasks. Therefore, it is of utmost importance to thoroughly study backdoor attacks using invisible triggers and develop effective mitigation strategies for diffusion models.
>
> **Q: Convergence concerns**
>
> A: Thanks for the feedback. We would like to further discuss the convergence of the optimization problem. We have empirically observed that the training process generally converges quickly, typically within approximately 50 epochs, across various settings, including different norm bounds, training epochs, and poisoning rates. However, we have also noticed that the training process can be slower when a very small norm bound is applied. This can be attributed to the increased difficulty in effectively injecting backdoors when the norm bound is set to a very small value.
>
> To provide more insights and guidance for practical implementation, we have included detailed ablation results on different norm bounds in Appendix H. These results can serve as a reference for selecting appropriate settings in practice.
>
> [a] Chou, Sheng-Yen, Pin-Yu Chen, and Tsung-Yi Ho. "How to backdoor diffusion models?." Proceedings of the IEEE/CVF Conference on Computer Vision and Pattern Recognition. 2023.
>
> [b] Baranchuk, Dmitry, et al. "Label-efficient semantic segmentation with diffusion models." arXiv preprint arXiv:2112.03126 (2021).
>
> [c] Chen, Huayu, et al. "Offline reinforcement learning via high-fidelity generative behavior modeling." arXiv preprint arXiv:2209.14548 (2022).
>
> [d] Chen, Shoufa, et al. "Diffusiondet: Diffusion model for object detection." Proceedings of the IEEE/CVF International Conference on Computer Vision. 2023.

---

### Official Review · Reviewer_ZqQk · 2023-10-31

**Soundness:** 2 fair
**Presentation:** 2 fair
**Contribution:** 2 fair
**Rating:** 3
**Confidence:** 4

**Summary:**

This paper proposes a novel optimization framework to inject invisible backdoors into diffusion models. Diffusion models are used for high-quality image generation but can also pose security threats. Current backdoor attacks on diffusion models rely on visible patterns that are easily detected. To address this, the authors propose a new framework to learn invisible triggers, making the inserted backdoor more stealthy and robust.
This framework can be applied to both unconditional and conditional diffusion models, with the latter being the first to show how to backdoor diffusion models in a text-guided image inpainting pipeline. The authors conduct experiments to verify the effectiveness and stealthiness of the proposed framework. This paper highlights the security risks associated with diffusion models and provides insights into mitigating these vulnerabilities through the use of invisible triggers.

**Strengths:**

1. The paper pioneers the study of backdoor attack problems for inpainting diffusion models.

**Weaknesses:**

1. The threat scenario presented seems impractical. Assuming that an attacker can control both the training process (using the bi-level training approach suggested in this work) and the usage of a diffusion model (specifically, choosing the noise used for inference) appears unrealistic in the context of diffusion models. If the attacker can dictate which noise the user employs for inference, the question of whether the trigger within the noise is visible or not becomes less significant.

2. The experiments conducted are insufficient. For instance, an ablation study for the crucial hyperparameter - the norm bound of the trigger - is absent.

3. I would like to encourage the authors to discuss the performance of the proposed attack when faced with defensive measures.

**Questions:**

1. What is the value of norm bound of trigger C actually used in the experiments?

---

> ### Author Response · Authors · 2023-11-22
> **Response to Reviewer ZqQk**
>
> Thanks a lot for the suggestions. We have addressed your concerns in the following.
>
> **Q: Threat scenario**
>
> A: Thanks for the feedback. We would like to provide further explanation and discussion on the attack setting and threat scenario. Similar with the setting in previous work[a], the attacker only has control over the training process. However, the attacker DOES NOT have the control over the choice of the initial noise which can be any noise randomly sampled from a Gaussian distribution. From Figure 1 and Figure 2, we could see the previous work needs the initial noise to have a designated pattern to make the backdoor activated and this could easily detected by human inspection and some distribution changed based detector, where the attacker will have little chance to activate the trigger. In contrast, our proposed method inserts invisible triggers into the initial noise. Hence the victim user won't be aware of the existence of the trigger, making the attack more stealthy and robust. For instance, in the unconditional case, we present two types of triggers, namely universal triggers and distribution-based triggers, and neither of them requires specific noise to be effective. This problem becomes more crucial in the conditional diffusion setting where the diffusion model is conditioned on various priors. If the priors are severely perturbed with an visible vision, it could be easily spotted.
>
> **Q: Missing ablation experiments**
>
> A: Sorry for the possible confusion. Because of the space limit, we have conducted detailed ablation studies in Appendix H of the original submitted supplementary material on the choice of different hyperparameters including the norm bound, poisoning rates. We also conduct more experiments on multiple trigger-target pairs, finetuning scheme, different samplers in the Appendix.
>
> **Q: Performance against defense**
>
> A: Sorry for the confusion. In the original submission, we have conducted experiments on defense methods. Specifically, in Section 4.4, we evaluate the performance of our framework when confronted with the defense method designed specifically for diffusion models as presented in [a]. We also include an additional defense method (ANP [b]) widely used in classification models in the revised manuscript and supplementary material. The results demonstrate that our proposed framework remains effective even in the presence of the defense. For more detailed information, please refer to Section 4.4(marked in blue) and Appendix G.
>
> **Q: Value of norm bound C**
>
> A: Sorry for the confusion. In Section 4.2, we have specified the norm bound $C$ used by stating "For CIFAR10,  $\ell_\infty$ norm bound is set as 0.2 ..." and "We show the results on high-resolution datasets CELEBA-HQ...., where the $\ell_\infty$ norm bound is also set as 0.2 ...". In Section 4.3, we have specified as "By setting the norm bound as 0.04, we show the quantitative results ...".
>
> [a] Chou, Sheng-Yen, Pin-Yu Chen, and Tsung-Yi Ho. "How to backdoor diffusion models?." Proceedings of the IEEE/CVF Conference on Computer Vision and Pattern Recognition. 2023.
>
> [b] Wu, Dongxian, and Yisen Wang. "Adversarial neuron pruning purifies backdoored deep models." Advances in Neural Information Processing Systems 34 (2021): 16913-16925.

---

### Official Review · Reviewer_hgfj · 2023-11-01

**Soundness:** 2 fair
**Presentation:** 3 good
**Contribution:** 2 fair
**Rating:** 3
**Confidence:** 2

**Summary:**

This paper focuses on developing a backdoor technique in diffusion models. the introduced technique is novel mainly because the designed trigger is invisible  (by human inspection). this work is the first to demonstrate how to backdoor diffusion models in text-guided image editing/inpainting pipelines. The results show successful attempt at implanting a backdoor and accessing it.

**Strengths:**

* this work is the first to demonstrate how to backdoor diffusion models in text-guided image editing/inpainting pipelines.
* the proposed attach works on inpainting tasks

**Weaknesses:**

the biggest weakness is the the backdoor is quite visible. When I zoom in on, say Fig 7 can I clearly see the images with trigger have been tempered with.

another weakness is the lack of comparison with other existing works

**Questions:**

why is the backdoor invisible when its' clearly visible? is it a special terminology people use in this field of work?

**Details Of Ethics Concerns:**

this paper exposes a way to implant backdoor in diffusion models. but did not offer a solution. the authors say "For future work, we will explore effective defense methods to mitigate possible backdoor in diffusion models.". I'm flagging this just to be on the safe side

---

> ### Author Response · Authors · 2023-11-22
> **Response to Reviewer hgfj**
>
> Thanks a lot for the suggestions. We have addressed your concerns in the following.
>
> **Q: Invisibility of the triggers**
>
> A: Thanks for your valuable comment. We would like to provide further explanation regarding the invisibility of the triggers. In the literature on attacks, the invisibility can be regulated by the norm bound, which corresponds to the constant C in Equation 8 of our manuscript. A smaller norm bound indicates more invisible triggers. The norm bound is set at 0.06 in Figure 7. In comparison, the results presented in Figure 6 utilize a smaller norm bound (0.04), rendering them more invisible than those in Figure 7. Consequently, one can anticipate that the triggers would exhibit even greater invisibility with a smaller norm bound. We also discussed this in Section 4.3. Moreover, in Appendix H, we conduct a detailed ablation study on different norm bounds for the unconditional case.
>
> The primary contribution of our work lies in proposing a unified framework for injecting invisible triggers into both unconditional and conditional diffusion models. In practice, the invisibility of the injected triggers can be controlled through the norm bound.
>
> **Q: Comparison with existing work**
>
> A: Thanks for the feedback. In our manuscript, we have discussed previous works that focus on backdooring diffusion models using visible triggers. However, to the best of our knowledge, no existing work has explored backdooring diffusion models with invisible triggers. And the backdoor methods on classification task could not simply extend to generative models because of different threat model, objective, training paradigm.

---

### Official Review · Reviewer_KSEM · 2023-11-01

**Soundness:** 2 fair
**Presentation:** 3 good
**Contribution:** 1 poor
**Rating:** 3
**Confidence:** 4

**Summary:**

The paper proposes a backdoor attack method on diffusion models. The proposed attack generates imperceptible triggers for both the unconditional (random noise input) and conditional (text input and masked image) cases. The trigger is essentially learned via a generator in a bi-level optimization approach, which alternates between learning the trigger and learning to inject the backdoor. The empirical results show that the proposed attack generates imperceptible triggers while maintaining the utility of the main task.

**Strengths:**

The paper has the following strengths:

1. The proposed attack seems to be capable of both conditional and unconditional cases, exposing the backdoor risks on training the diffusion models.
2. The experimental results show the effectiveness of the proposed methods, specifically the intended imperceptibility.

**Weaknesses:**

While the paper solves an important problem for understanding the security risks of using diffusion models, there are some major weaknesses:

1. Achieving imperceptibility for the unconditional case is not well-motivated. For example, even when having a small residual, the random noise generated by this method and of previous works are still not recognizable.

2. Another concern is that, while the adversary in backdooring classification can benefit from targeted, incorrect prediction (e.g., turn a decision of rejecting a credit application into an accepting one), it's unclear what is the benefit of backdooring diffusion model, since the adversary is the one who receives and uses the generated images for their own benefits. In order words, the paper lacks a discussion on why the adversary wants to attack themselves.

3. The novelty of the technique is trivial. It is merely an application of previous bi-level optimization works developed for backdooring classification models (e.g.,  Doan et al. ICCV 2021, Doan et al. NeurIPS 2021), but replacing the classifier with a diffusion model. However, these works in the classification backdoor are not mentioned at al, which make the technical contributions of the paper very limited.

4. Limited evaluation of defense techniques. In backdooring classification, there are several defenses that rely on input perturbation, and I think that their evaluation on the proposed method is essential for a comprehensive understanding of the attacks.

Doan et al. Lira: Learnable, imperceptible and robust backdoor attacks. ICCV 2021.
Doan et al. Backdoor attack with imperceptible input and latent modification. NeurIPS 2021.

**Questions:**

Please see the comments on weaknesses.

---

> ### Author Response · Authors · 2023-11-22
> **Response to Reviewer KSEM (1/2)**
>
> Thanks a lot for the valuable suggestions. We have addressed your concerns in the following.
>
> **Q: Motivation of invisible trigger for unconditional case**
>
> A: Thanks for your valuable comment. We would like to elaborate more on the importance and motivation of our work and have incorporated the discussion in the section Introduction(marked in blue) and Appendix B in the revised manuscript and supplementary material. As both mentioned in [e, f], it is important to improve the fidelity of poisoned examples that are used to inject the backdoor and hence reduce the perceptual detectability by human observers. In the unconditional case, it is thus important to make the sampled noise to be similar with random noise used in the practice or it could be easily filtered by human inspection. As shown in Figure 1 and Figure 2 in the manuscript, the triggers used by previous works (also in the unconditional case) could be easily detected through human inspection without any effort. In contrast, our proposed invisible trigger is nearly visually indistinguishable from the original input, which greatly increase attack's stealth so that human inspection would no longer effective.
>
> Moreover, we want to address our work is the first framework to propose a unified framework for learning invisible triggers to backdoor on not only unconditional but also conditional diffusion models. In addition to unconditional generation, invisible triggers are particularly practical in conditional diffusion models, which hasn't been explored and discussed by the previous works.
>
> **Q: Discussion on the benefit of backdooring diffusion models**
>
> A: Thanks for the valuable suggestions. We believe the backdoor attack in diffusion model is as important as in the classification model. Since the powerful models like Stable Diffusion is open-sourced, anyone could download the model and conduct malicious fine-tuning to insert a secret backdoor that can exhibit a designated action (e.g. generating a inappropriate or incorrect images). Explicitly, the generated output will be directly controlled by activating backdoor for conducting some bad actions like disseminating propaganda, generating fake contents etc.
> Meanwhile, implicitly, as also discussed in [a], the diffusion model has been widely in a lot of different downstream tasks and applications such as reinforcement learning, object detection, and semantic segmentation [b, c, d]. Hence if the diffusion model is backdoored, this Trojan effect can bring immeasurable cartographic damage to all downstream tasks and applications.
>
> We have incorporated the discussion in the section Introduction(marked in blue) and Appendix B in the revised manuscript and supplementary material. Please refer to Appendix B for the discussion.
>
> **Q: Novelty of the method**
>
> A: Thanks for pointing these valuable related works. We provide further discussion regarding the previous works on backdooring classification models through bi-level optimization and have incorporated the discussion into the revised manuscript and supplementary material. Please refer to the section Introduction(marked in blue) and Appendix C. Although previous works [e, f] also utilize a bi-level optimization to achieve a similar objective, we believe that learning an invisible backdoor trigger in the context of diffusion models is totally different and much harder than finding one in the classification model. The method developed for backdooring classification models cannot be directly or easily extended to backdoor diffusion models.  Specifically, the threat model is totally different. Diffusion models consist of diffusion and reverse processes that fundamentally differs from classification models. Backdooring diffusion model needs to have careful control of the training procedure while only poisoning data needs to be added in the classification model. At the same time, it is nontrivial and challenging to design the backdoor objective in the conditional and unconditional diffusion model  while it is relatively a simple task in the classification. To learn invisible backdoors for both unconditional and conditional diffusion models, the entire pipeline, training paradigm, and training loss have to be redesigned  to differ significantly when applying bi-level optimization to backdoor diffusion models.
> In this setting, the training loss, training paradigm, and pipeline are specifically designed based on the properties of diffusion models differing substantially from backdooring classification models through bi-level optimization.

---

> > ### Author Response · Authors · 2023-11-22
> > **Response to Reviewer KSEM (2/2)**
> >
> > **Q: Evaluation on defense methods**
> >
> > A: Thanks for your valuable advice. However, we believe it is nontrivial to extend the defenses in classification model into diffusion model since a dramatic difference in every aspects including the entire training paradigm, training loss and the objective. Even with modification, those defenses will easily fail. For example, following Baddiffusion [a], we adapt a widely used defense Adversarial Neuron Pruning (ANP) in classification to defense against our attack [a] by adding noise to the weights of our backdoored model. However, we find ANP performs even worse in our proposed attack, compared to the performance in [a], which is totally ineffective to our attack. The perturbated model by ANP immediately collapsed to a meaningless image or a black image. For detailed visualization results and quantitative results, please refer to Section 4.4(marked in blue) and Appendix G in the revised manuscript and supplementary material.
> >
> > In addition, in Section 4.4 in the manuscript, we also evaluate the defense method specifically designed and effective for diffusion models in [a] and highlight its ineffectiveness against the proposed backdoor attack. Please refer to Appendix G for more detailed results. To the best of our knowledge, there is currently no other defense method designed specifically for mitigating backdoors in diffusion models.
> >
> > [a] Chou, Sheng-Yen, Pin-Yu Chen, and Tsung-Yi Ho. "How to backdoor diffusion models?." Proceedings of the IEEE/CVF Conference on Computer Vision and Pattern Recognition. 2023.
> >
> > [b] Baranchuk, Dmitry, et al. "Label-efficient semantic segmentation with diffusion models." arXiv preprint arXiv:2112.03126 (2021).
> >
> > [c] Chen, Huayu, et al. "Offline reinforcement learning via high-fidelity generative behavior modeling." arXiv preprint arXiv:2209.14548 (2022).
> >
> > [d] Chen, Shoufa, et al. "Diffusiondet: Diffusion model for object detection." Proceedings of the IEEE/CVF International Conference on Computer Vision. 2023.
> >
> >
> > [e] Doan, Khoa, et al. "Lira: Learnable, imperceptible and robust backdoor attacks." Proceedings of the IEEE/CVF international conference on computer vision. 2021.
> >
> > [f] Doan, Khoa, Yingjie Lao, and Ping Li. "Backdoor attack with imperceptible input and latent modification." Advances in Neural Information Processing Systems 34 (2021): 18944-18957.

---

### Meta-Review · Area_Chair_w1ZM · 2023-12-05

**Metareview:**

The authors propose a new invisible backdoor attack for diffusion models. However, the scenario for the backdoor attacks is unclear which is also agreed by KSEM and ZqQK. Since the attacker controls the training but can't influence inference according to their response to ZqQK, the sampling noise can just be sampled as Gaussians by benign users. If the initial noise cannot be controlled by benign users, why we need invisible is still not clear.
Apart from their invisible settings, the paper shows no advantages over than former ones and the paper's performance is a little worse than the previous backdoor papers when evaluating the FID. Therefore, I decide to reject this paper.

**Justification For Why Not Higher Score:**

The paper's settings are not practical and the performance is not good.

**Justification For Why Not Lower Score:**

N/A

---

### Decision · Program_Chairs · 2024-01-16

Reject